# Movement of Chronic Wasting Disease Prions in Prairie, Boreal and Alpine Soils

**DOI:** 10.3390/pathogens12020269

**Published:** 2023-02-07

**Authors:** Alsu Kuznetsova, Debbie McKenzie, Bjørnar Ytrehus, Kjersti Selstad Utaaker, Judd M. Aiken

**Affiliations:** 1Department of Renewable Resources, University of Alberta, Edmonton, AB T6G 2G8, Canada; 2Centre for Prions and Protein Folding Diseases, University of Alberta, Edmonton, AB T6G 2M8, Canada; 3Department of Biological Sciences, University of Alberta, Edmonton, AB T6G 2M8, Canada; 4Norwegian Institute for Nature Research (NINA), 7034 Trondheim, Norway; 5Department of Biomedicine and Veterinary Public Health Sciences, Swedish University of Agricultural Sciences, SE-750 07 Uppsala, Sweden; 6Faculty of Biosciences and Aquaculture, Nord University, 8026 Bodø, Norway; 7Department of Agricultural, Food and Nutritional Sciences, University of Alberta, Edmonton, AB T6G 2M8, Canada

**Keywords:** chronic wasting disease, soil binding capacity, prions, soil columns

## Abstract

Chronic wasting disease (CWD) is a transmissible spongiform encephalopathy negatively impacting cervids on three continents. Soil can serve as a reservoir for horizontal transmission of CWD by interaction with the infectious prion protein (PrP^CWD^) shed by diseased individuals and from infected carcasses. We investigated the pathways for PrP^CWD^ migration in soil profiles using lab-scale soil columns, comparing PrP^CWD^ migration through pure soil minerals (quartz, illite and montmorillonite), and diverse soils from boreal (Luvisol, Brunisol) and prairie (Chernozem) regions. We analyzed the leachate of the soil columns by immunoblot and protein misfolding cyclic amplification (PMCA) and detected PrP in the leachates of columns composed of quartz, illite, Luvisol and Brunisol. Animal bioassay confirmed the presence of CWD infectivity in the leachates from quartz, illite and Luvisol columns. Leachates from columns with montmorillonite and prairie Chernozems did not contain PrP detectable by immunoblotting or PMCA; bioassay confirmed that the Chernozemic leachate was not infectious. Analysis of the solid phase of the columns confirmed the migration of PrP to lower layers in the illite column, while the strongest signal in the montmorillonite column remained close to the surface. Montmorillonite, the prevalent clay mineral in prairie soils, has the strongest prion binding ability; by contrast, illite, the main clay mineral in northern boreal and tundra soils, does not bind prions significantly. This suggests that in soils of North American CWD-endemic regions (Chernozems), PrP^CWD^ would remain on the soil surface due to avid binding to montmorillonite. In boreal Luvisols and mountain Brunisols, prions that pass through the leaf litter will continue to move through the soil mineral horizon, becoming less bioavailable. In light-textured soils where quartz is a dominant mineral, the majority of the infectious prions will move through the soil profile. Local soil properties may consequently determine the efficiency of environmental transmission of CWD.

## 1. Introduction

Chronic wasting disease (CWD) is a fatal, contagious prion disease affecting free-ranging white-tailed deer, mule deer, elk, and moose, as well as farmed cervids in North America. CWD is spreading throughout North America (Figure 1), and there has been a recent outbreak in reindeer and moose in Scandinavia [1] and in several deer species in South Korea [2]. The North American geographic distribution of CWD in deer and elk has been expanding and will ultimately result in the exposure of caribou to CWD. The physical distance between caribou (*Rangifer tarandus*, the same species as reindeer) and the CWD-endemic region appears to be the only factor limiting the exposure and potential transmission of CWD to caribou herds in North America (Figure 1).

A remarkable property of infectious prions is their persistence in external environments, especially their ability to remain infective for years to decades [3,4]. Soils may act as a natural environmental reservoir, contributing to the transmission of CWD [5,6,7] due to the direct deposition and persistence of infectious prions in soils [3,8,9]. Variations in soil properties affect prion persistence and transmission [5,10,11], as prions associated with natural soils could remain near the soil surface, making them available for ingestion by grazing animals. Deer consume significant quantities of soil associated with the roots of leafy plants, especially during winter foraging [12,13]. In addition, cervids also ingest soil intentionally to supplement mineral intake [12,14]. Part of the emerging picture of prion stability in soil involves avid binding to minerals and interaction with other soil compounds. Montmorillonite (mte) forms an especially strong association with PrP^Sc^, and prion infectivity may be increased by the prion-mineral interactions that occur during passage through soil [8]. It is hypothesized that the mechanism of mineral binding involves electrostatic attractions between the positively charged N-terminal domain of the prion protein (which is rich in lysine, histidine, and arginine) and the negatively charged clay minerals [15]. In addition, organic polyanions (e.g., humic acids as part of soil organic matter) [10,16,17], some metals and their oxides [18,19,20,21,22], and other soil components [23] have the capacity to interact with prions, influencing their bioavailability and persistence in the environment. The mechanism of how prions and soil organic matter (SOM) interact is unknown; however, SOM can interfere with detection of infectious prions [6,16,24]. We hypothesize that the fate of prions in soils from different regions varies due to differences in soil properties, including mineralogical composition, SOM content, texture, and their prion binding ability.

CWD endemic regions in North America are generally associated with clay, loamy, mte soils with high amounts of humus [6]. Such soils are most prevalent in CWD-endemic regions, but less common in northern regions of North America and Europe where quartz–illite sandy soils with low amounts of humus content prevail (Luvisols and Podzols, Figure 1 and Appendix A). CWD transmission among cervids in the North America CWD-endemic region was enhanced in regions with relatively high soil-clay content [11]. The diverse mineralogical composition of these soils may affect prion binding to soils, as well as the soil’s ability to preserve and transport prions.

Soils of different regions also vary in their moisture regime, and this, together with the binding capacity of soil, likely impacts PrP^CWD^ migration in soils. Soils from the prairie regions (Chernozems) have either an ustic or aridic regime. This means that the moisture in the soil is limited because the stored moisture plus rainfall is less than the amount of water lost via evapotranspiration. Soils from boreal and tundra regions (Luvisols, Podzols, Brunisols and Leptosols) have either an ustic or udic moisture regime, in which the amount of stored moisture plus rainfall is equal to, or exceeds, evapotranspiration. Water moves through the soil profile of boreal and tundra soils in most years, while it does not in prairie soils. Considering the soil moisture regime may affect the movement of prions of prions from surface to lower horizons, PrP^CWD^ may not adhere to the soil surface in boreal soils. Little is known regarding PrP^CWD^ movement through natural soils in environmental conditions; we speculate that fine textured soils with high humus content would trap prions near the surface, whereas coarse-textured soils with non-mte mineralogical composition might increase prion mobility.

In the natural environment, there are four main scenarios for the fate of the prions in soils: (i) prions interact with the surface soil organic or mineral horizons and remain bioavailable for animals; (ii) prions become bound to soil substances that decrease bioavailability; (iii) prions are transported into lower soil horizons and are unavailable for consumption; or (iv) prions migrate through the whole soil profile. We hypothesized that infectious prions shed onto soil surfaces would, dependent upon soil type, exhibit different migration patterns through soils. Our findings suggest that soil texture, mineralogy, humus content, pore structure and related soil properties deserve greater attention in assessing risks of CWD transmission and detailed investigation of environmental factors have potential implications for both Europe and North America.

## 2. Materials and Methods

### 2.1. Soils and Minerals

Migration of PrP was investigated in six surface soil horizons of four different soils: horizons LFH (surface plant litter: Litter, Fermented, Humic) and Ae (upper eluvial mineral) of Luvisol, horizons LFH and Bf (iron-enriched illuvial) of Brunisol and Ah (near surface mineral containing accumulated humic material) horizons of two Chernozemic soils. These soil samples were collected in Alberta, Canada and represent soil cover of boreal, tundra and prairie regions. Luvisols and Brunisols are developed in boreal and tundra ecozones, the Chernozems are the dominant types in the prairies (Appendix A). Leptosol, for the binding experiment, was collected in the alpine region (Nordfjella, Norway). Soil properties are summarized in Table 1. The minerals used for binding and column experiments were purchased from Ward’s Science: quartz, illite and mte.

### 2.2. Prion Preparation, Binding Experiments and Immunoblotting

CWD prion isolates/homogenates were from CWD prion-infected brain tissue of transgenic mice expressing elk PrP (TgElk) or tg33 mice, which express wild-type white-tailed deer (WTD) PrP (tg33) and hunter-harvested CWD-infected mule deer. Uninfected brain tissue from transgenic tg33 and tgElk mice served as controls. Brains were homogenized (10% *w*/*v*) in water, and clarified at 800 *g* for 5 min when applicable [5]. In binding experiments, identical amounts of 10% brain homogenate (BH^CWD^ or uninfected: NBH) were incubated with soil minerals, soil horizons or water (control) at 4 °C for 24 h. Samples were fractionated through a sucrose cushion (1M) to separate bound and unbound prions. The resulting pellet, containing bound prions, and the methanol-precipitated supernatant, containing unbound prions, were resuspended in 40 µL of 5 × SDS sample buffer and heated at 100 °C for 10 min. Samples were analyzed by Western blot. Each sample was resolved on 15-well 12% NuPAGE bis-Tris gels (Invitrogen), transferred to polyvinylidene difluoride membrane and probed with an anti-PrP antibody (Bar 224 at 1:20,000). Proteinase-resistant PrP^res^ was identified by digestion of 100 µL of sample with 3.5 µg of Proteinase K (PK) (Roche) for 45 min at 37 °C in a volume of 70 µL (50 mg/mL PK final concentration). Digestion was terminated by addition of 30 µL of 5 × SDS sample buffer and heating at 100 °C for 10 min.

### 2.3. PrP^CWD^ Transport Experiments

The mineral/soil columns were set up in 50 mL (24 mm internal diameter, 100 mm high) polypropylene tubes (Corning, #430829) with perforated bottoms (Appendix A). The columns were filled with 40 mL of minerals (quartz (Qz), illite (Ill), and montmorillonite (mte)) or soil horizons (the LFH and Ae horizons of Luvisol and the LFH and Bf horizons of Brunisol from the boreal region; and the Ah horizons of two different Chernozems from the prairie region (Table 2)). To improve the penetrability of the clays (mte and illite), quartz sand was added in the ratio of 3:7 for columns incubated for 6 weeks, or in the ratio of 1:1 for columns incubated 3 weeks, mimicking sandy loam texture. Minerals and soils were gently packed into columns without any tamping to preserve the initial soil structure and pore space. Brain homogenate (BH, 0.4 mL, tg33 mice) was applied to the surface. The columns were irrigated weekly with 12 mL of DI water, simulating the 90 mm/month precipitation common to the boreal region of Western Canada in the summer months. Leachates were collected the same day to analyze the presence of PrP. To obtain a minimum of 5 mL leachate, a vacuum was applied to the columns as needed.

After the final irrigation, the solid phase was fractionated into 2 mL layers. PrP^CWD^ was extracted by boiling in SDS buffer for 10 min. The sample was centrifuged, and the supernatant transferred in a new tube and the PrP methanol precipitated. The pellet was resuspended in SDS buffer and boiled at 100 °C for 10 min. The samples were analyzed by Western blot as described above.

### 2.4. Protein Misfolding Cyclic Amplification (PMCA) Assay

PMCA was used to identify PrP^CWD^ in leachates obtained from the columns. Perfused tg33 BH was used as a substrate. An amplification control of 10% CWD-infected brain homogenate samples was serially diluted 10-fold in 10% non-infective BH. Subsequently, 90 μL of each 10-fold dilution series and a negative control (10% uninfected BH) were placed in a QSonica Q700 sonicator (Misonix Inc., Farmingdale, NY, USA). Samples were incubated at 37 °C and subjected to a single round of amplification with 144 cycles of 30 s sonication followed by 30 min incubation. Identical samples prepared for PMCA were incubated at 37 °C for the same period of time without sonication as a non-PMCA control. For PrP^CWD^, two rounds of PMCA were performed under the same conditions, with the second round using a 1:10 dilution of amplified materials from the previous round in 10% uninfected tg33 BH substrate. All PMCA products and non-PMCA controls were PK digested and analyzed by Western blot using mAb Bar224 (1:20,000).

### 2.5. Infectivity Bioassay

Leachates obtained at the second week from the Qz, illite, Luvisolic Ae horizon and Chernozem columns were pasteurized 10 min at 80 °C and 30 µL intracerebrally inoculated into tgElk mice. The PrP^CWD^ starting material (0.1%, 0.01%, 0.001% BH) and uninfected BH (1%) was inoculated into control animals (tgElk mice). Mice were monitored daily for the onset of clinical symptoms and euthanized upon established clinical disease. Brains were analyzed for protease-resistant PrP^res^ by protease digestion followed by immunoblotting.

### 2.6. Ethics Statement

All work with animals was performed in compliance with the Canadian Council on Animal Care Guidelines and Policies. All procedures involving animals were reviewed and approved by the Health Sciences Animal Care and Use Committee of the University of Alberta under protocol “Etiology and Pathogenesis of Prion Diseases” AUP # 914.

## 3. Results and Discussion

### 3.1. PrP^CWD^ Interaction with Soil Minerals and Soils

More than 10 different types of soils are present in, and neighboring, the North American CWD-endemic region, and there are at least three main types in Scandinavia where CWD has been detected. The soils vary in texture, pH, mineralogy, SOM composition, humus content and other properties. To estimate the impact of these differences on soil prion transport, profiles of the diverse soils from CWD-endemic and adjacent regions were analyzed (Appendix A).

After performing basic soil analyses (Table 1), we focused on two Chernozemic soils that represent the North American CWD-endemic region, as well as a Luvisolic soil and a Brunisolic soil from the boreal region (Table 1, Figure 1), and Leptosol soil from the alpine region in Norway. The surface Ah horizons of Chernozems both have clay loamy and loamy texture, and high total organic carbon (TOC) and HA concentrations, 3.9–4% and 15–19 g L^−1^, respectively. In the Luvisolic soil, two surface horizons were analyzed, organic litter LFH and mineral podzolic Ae. Both horizons have a slightly acidic pH, but differ in their total carbon content, in LFH, TOC reached 40%, while it was much lower in the podzolic Ae horizon (~1.1%). In the Brunisolic soil surface horizons, the organic litter LF and mineral Bf were also analyzed. The TOC content in the LF horizon reached 38% due to the high amount of plant debris, while the TOC was quite low (1%) in the Bf horizon. The mineralogy of the clay fractions, which has particles smaller than 2 µm, varied. In the Chernozems, mte and kaolinite prevailed, while mica and illite were dominant in the Luvisols and Brunisols clay fractions. The Leptosol was collected in the Nordfjella region in Norway, where an outbreak of CWD in wild reindeer was discovered in 2016. The sample was collected in reindeer habitat at 1319 m above sea level in an area where the thin soil layer is only sparsely covered with dwarf shrubs, bryophytes and lichens. The soil was characterized by a slightly acidic silt sandy mineral horizon with very low clay content and vermiculite–quartz mineralogy.

The total prion binding capacity of soils is related to the binding capacity of its individual components. To better understand this, we first compared the prion binding capacity of pure soil minerals, as soils from the prairie and boreal regions have differing mineral content and the solid phase comprises more than half of the soil matrix. Using the detection method developed for determining bound and unbound prions [5], we estimated the binding capacity of different soil minerals. We used mte, whose strong binding capacity has been well-characterized [5], as a control to estimate the relative binding capacity of illite, which has not been described.

Brain homogenates from CWD-infected mule deer were tested for mineral binding. Prions from clarified BH (where cell debris were removed during the clarification process) were present predominately in the supernatant, and did not pellet after centrifugation through a sucrose cushion (800× *g*, 10 min) (Figure 2). Incubation with quartz and Leptosol resulted in PrP remaining in the supernatant, i.e., a negligible amount of PrP bound to quartz or Leptosol. Incubation of BH with mte as well as Chernozemic soil showed that almost all PrP was in the pellet (bound prions), while PrP was undetectable in the supernatant (unbound prions). When incubated with illite, the majority of prions remain in the supernatant after centrifugation. The lower total recovery of PrP^CWD^ from soils compared to minerals might be related to other soil compounds (i.e., soil organic matter) that prevented desorption of PrP^CWD^. Our results indicate that PrP^CWD^ were entirely bound to mte and Chernozem, while the binding capacity of illite, in comparison, was significantly reduced, while Leptosol and quartz had negligible ability to bind prions.

One hypothesis on the mechanism by which organic matter (including proteins) binds to soil minerals mainly involves surface complexation where organic molecules adhere to and stabilize on mineral surfaces using functional groups (e.g., carboxyl, hydroxyl, phenolic) [25]. This hypothesis is not, however, compatible with the variation in binding capacity of mte and illite, as both have a negative surface charge. We hypothesize that differences in their crystalline structure and physical properties may contribute to the diversity of binding capacity. Both illite and mte have a 2:1 layer structure. However, in illite the interlayer space is fixed at 0.3 nm, while it can be expanded up to 4 nm in mte [26]. The mte formula is often expressed as Al_2_O_3_ × 4SiO_2_ × H_2_O × M^+^, where M^+^ is an interlayer cation which neutralizes the negative charge of the minerals. The specific surface area of mte is approximately 700 to 800 m^2^ × g^−1^, and, as the component layers are not strongly bonded, the mineral can exhibit interlayer swelling, causing the volume of the clay to double. The negative charge of mte arises mainly from an isomorphous substitution, and thus the charge density or distribution of electrostatic charge on proteins will influence their adsorption by mte. The adsorption of organic compounds by clay surfaces leads to the formation of organomineral complexes and an interaction can occur not only on the outer surfaces, but X-ray diffraction analysis of basal spacings of clays has shown that a considerable amount of the organic molecules can penetrate the intermicellar regions of 2:1 expanding types of clays, and organic ions are able to replace inorganic cations in the interlayer position [26]. It is possible that the positively charged N-terminal domain of the prion protein may partially enter into the mte interlayer position and be strongly fixed there. A 20 kDa protein mass is equivalent to 1.78 nm (min. radius) [27], thus PrP might vary from 1.78 to 2.1 nm (37 kDa). The unglycosolated form of PrP is, therefore, smaller than the inter-layer space of mte, and some peripheral part of molecule could be adsorbed between the clay layers. The interlayer binding can explain the avid binding and strong association of PrP to mte. The illite minerals are micaceous types of clay and have a secondary origin from muscovite. It is believed that a continuous series of illite species exists between muscovite and smectite and mixed layering of illite-smectite often occurs. The illite unit cell formula is ((OH)_2_(Si_3.3_Al_0.7_)(Al_1.33_Mg_0.4_Fe_0.3_)O_10_)_−1.3_ 1.3K^+^ [26]. Illite contains interlayer potassium; thus, the unit layers are bonded more strongly and the intermicellar spaces of illite do not expand upon addition of water; therefore, interlayers would not accommodate the PrP^CWD^ and bind it avidly. The above evidence suggests that the physical properties of illite prevent significant binding of prions in comparison to the binding by mte.

### 3.2. PrP^CWD^ Transport Experiments

The variability in the binding capacities of soil minerals can significantly impact the prion binding capacity of whole soils. The strong electrostatic attraction between minerals and proteins would account for our observation that prions remain at the contaminated mte soil surface with negligible leaching to lower soil horizons. Mte is a dominant clay mineral in prairie soils, while illite and Qz are prevalent in the clay fraction of the boreal and tundra soils [28]. Differences in mineral binding capacity and soil textures, as well as variability in soil organic content, may impact the mobility and migration of prions through different soil profiles.

We tested two hypotheses: (1) in mte-rich soils, prions will remain in the surface soil horizon; and (2) in illite-rich soils, prions will remain unbound and can move into lower soil horizons. Experiments varying in length of washing were performed using soil columns that included different soil minerals or individual soil horizons. In one experiment, CWD prions were layered over three pure minerals (quartz, illite and mte), watered regularly for 6 weeks. In the second experiment, prions were layered over three pure minerals (quartz, illite and mte), and six pristine soil horizons (northern Luvisol, horizons LFH and Ae; mountain Brunisol, horizons LFH and Bf; 2 Chernozems Ah horizons) and examined for 3 weeks (Appendix A). During these experiments, we tested not only the binding capacity of soil compounds, but also the ability of prions to migrate through the soil profile.

#### 3.2.1. Leachate from Mineral/Soil Column Experiment

The initial mineral columns used pure soil minerals to determine how minerals with different binding capacities alter prion migration. Brain homogenates from CWD-infected or -uninfected (control) animals were applied to the surface of the columns. Precipitation, which induces water downdrafts in the soil profile and the real moisture regime of the soil, was modeled by weekly watering. Leachates were collected from the bottom of the columns and analyzed for PrP using immunoblotting (Table 2, Appendix A). PrP from quartz columns with the infected or uninfected BH was detectable in the leachate after the first week, indicating that both PrP^C^ and PrP^CWD^ migrated through the quartz columns (Figure 3). PrP continued to be detected in the quartz column leachate for the next 4 weeks with the intensity of signal decreasing from week to week. PrP^CWD^ was detectable until the 5th week of irrigation, while PrP^C^ was detectable in the leachate only during the first two weeks of irrigation. For the other mineral columns (illite, mte), the PrP was not detectable in leachates by immunoblot throughout the 6 weeks (Figure 3).

For the whole soils, PrP^CWD^ migrated through the organic (LFH-type) horizon of Luvisolic soils and the pure quartz during the first week (Figure 4), but was only detected in the quartz column leachate during the second (Figure 4) and third weeks of the experiment (Appendix A). From the columns containing mte, illite, the mineral horizons of Luvisolic and Brunisolic soils, and the organic (Ah-type) horizons of Chernozems, conversely, no PrP signal was present in the leachates during the three weeks of irrigation (Figure 4). Although PrP was detectable in both leachates from the quartz and LFH-horizons columns, the mechanism of the water transport obviously differ. The LFH horizon is a non-homogeneous mixture of differentially degraded porous plant debris. Therefore, water will migrate through the column using these pores instead of moisturizing the whole volume evenly and penetrating the solid phase as a continuous front, as it would in the quartz mineral column. Furthermore, brain homogenate applied to the column surface will be distributed unevenly in these horizons, resulting in the sequestering of some portion of BH away from preferable water pathways, and therefore remaining trapped in the column at the second (Figure 4) and third weeks.

To determine whether the leachates from the Chernozem column contained low levels of PrP^CWD^, we analyzed select samples using PMCA (Figure 5). PMCA is a highly sensitive means of amplifying and detecting infectious prions that increases sensitivity 10,000-fold for prion detection in soil compared to Western blot [29]. We analyzed leachates from the quartz, illite, Luvisol Ae, Brunisol Bf and Chernozem 1 columns collected at 3 weeks. A serial dilution of 1% CWD BH, the same BH applied to the columns, was included as a control. The control was detectable at dilutions to 10^−4^ after a single round of PMCA, and at dilutions up to 10^−6^ after a second round of PMCA (Figure 5). PrP^CWD^ was detected in the quartz, illite, Luvisol and Brunisol column leachates after the first and second rounds of PMCA. In the leachate obtained from the Chernozem column, PrP^CWD^ was not detectable after the first and second rounds of PMCA. These data confirm that PrP^CWD^ is transported through n quartz, illite, Luvisol Ae and Brunisol Bf columns; while binding to Chernozem soil compounds, PrP^CWD^ was bound to the Chernozem soil compounds, preventing the transportation of CWD-prions through the column and into the leachate.

We tested the infectivity of the leachates using tgElk mice intracerebrally inoculated with leachate aliquots from the quartz, illite, Luvisolic Ae and Chernozem columns. All mice inoculated with CWD-infected BH (positive control) showed clinical signs with incubation periods of 126 ± 9 dpi (0.1% BH) to 158 ± 7 dpi (0.001% BH), all accumulated PrP^res^ in their brains (Figure 6). Half of the mice (6 out of 11) for leachates from quartz and Luvisolic Ae columns exhibited clinical signs after 170 dpi and 183 dpi, respectively, and were positive for PrP^res^ by Western blot. Three of eight mice inoculated with illite column leachate showed clinical signs (incubation period 170 dpi). None of the mice (*n* = 8) inoculated with leachate from Chernozem columns showed clinical signs of disease. All remaining mice were euthanized at 225 dpi without clinical signs and accumulation of PrP^res^ in their brains. Bioassay data clearly showed that leachate collected from quartz, illite and Luvisolic Ae columns contained infective PrP^CWD^, while leachate from the Chernozem column was not infective.

In summary, we found PrP in leachates from quartz, and the upper soil organic horizon (LFH) of boreal Luvisolic and Brunisolic soil columns. Although the immunoblot was negative for PrP for the illite, and the mineral horizons of Luvisol and Brunisol leachates, PMCA and infectivity bioassays confirmed PrP^CWD^ in these leachates. Notably, the leachate from Chernozem soil, a common soil of the CWD-endemic region of North America, was not infective.

#### 3.2.2. Extraction from Solids

To further investigate the prion migration in soil profiles, we extracted PrP from solids (Section 2). After 3 weeks of column rinsing, the solid phase was subdivided into layers, and each layer extracted and analyzed by immunoblot (Appendix A). Each layer represents 0.5–2 mL of solids. Varying PrP signal was detected in different layers of mineral columns (Appendix A). As expected, the majority of the prions were washed out of the quartz column—PrP was not detected in any layer (Appendix A). Prions migrated out of the upper layers of the illite column with the PrP present in the middle of the column (layers 7–15), indicating that prion migration into the lower layers also occurred but less than in the quartz column. In the mte column, the strongest signal was detected close to the column surface in layers 1–3, with the intensity of signal decreasing with depth, disappearing completely below layers 7–10 (Appendix A). As expected, the PrP signal was altered after extraction from the mte column, as the 35 kDa band became truncated, with only 20–25 kDa being visible on the immunoblot [5,8].

Previous studies examining infectious prion migration through soils have contradictory results, in some, PrP migration was observed, while in others, prions were retained near the point of initial loading [3,30,31]. Brown and Gadjusek (1991) recovered infectivity from TSE-infected hamster brain material buried in garden soil for 3 years and noted a minimal vertical movement of the infectious agent. Migration of PrP^TSE^ was not detected through different landfill materials [31]. Similarly, no significant lateral movement of infectivity was observed during the 5-year period in either a clay or sandy soil [32]. The discrepancy between the obtained results may be explained by different properties of the soils studied. The interaction of proteins with soil minerals can change protein structure, causing slight unfolding, aggregation or disaggregation [33]. As PrP structure impacts prion transmissibility and pathogenesis [34,35], it is possible that soil binding could affect these properties. At the same time, in the environment, PrP competes for sorption sites with thousands of other constituents, likely slowing PrP movement through the soil surface and inhibiting initial binding to soil components. Therefore, to predict the fate of prions in soils, it is important to analyze soil properties [36]. Prion persistence in soil is a dynamic balance between protective microenvironments, such as binding to minerals, which favor prion stability [37,38] versus processes leading to their degradation.

Our results suggest that the migration of PrP^CWD^ through soil will be minimal in natural soils with a silt loam/clay loam texture, mte mineralogical composition and high HA content (Figure 7). The majority of PrP^CWD^ shed into these soils will remain at or near the location of deposition and be bioavailable. However, soils with a sandy texture and illite or quartz mineralogical composition will facilitate movement of prions into lower soil horizons due to the lower prion binding capacity of illite and quartz compared to mte.

## 4. Conclusions

Our data suggest that CWD infectivity will remain on the surface of soils of temperate regions, including the Chernozems of Northern America and Cambisols of Europe. Migration of CWD prions deeper into the soil profile is expected to be restricted to finely textured soils with relatively high humus content. However, soils with a sandy-loam texture and illite–quartz minerology, such as the Luvisols, Podzols, Brunisols and Leptosols of boreal and tundra regions, will likely transport the majority of prions through the upper mineral soil horizons and into the lower horizons, becoming less bioavailable. In sandy-textured soils with low humus content (for example, the Podzols, the dominant soils found along the American north-west and north-east coast, Scandinavia, and Northern Europe), the majority of prions would likely be transported to the deepest soil horizons. Overall, mineral horizons of sandy soils of the boreal and tundra regions are less favorable to the horizontal transmission of CWD; these soil horizons make PrP less bioavailable due to prion transport from superficial to lower horizons. This suggests that, in soils of North American CWD-endemic regions (prairie Chernozems), CWD prions would remain on the soil surface due to strong binding to montmorillonite, while in boreal Luvisols and mountain Brunisols, some prions will move through the plant litter LFH, and partially through the upper soil mineral horizon into lower horizons. In the dominating, natural light-textured soils of the CWD-affected region in Scandinavia, where quartz is a dominant mineral, our results suggest that the majority of prions will move through the mineral soil profile. However, how prions interact with the surface plant litter soil horizon of northern soils is still unclear and needs more detailed studies. These studies indicate some binding of CWD prions to leaf litter suggesting the retention and bioavailability of infectivity in the Scandinavian soils examined. In addition, anthropogenic activity may change the properties of soil and thereby influence prion dynamics in the environment (work in progress).

Consequently, environmental transmission of CWD via soil is facilitated in the prairie regions, while the lowered bioavailability of environmental CWD prions in boreal, tundra and alpine surface soils may reduce the efficiency of transmission.

Future studies of the fate of prions in the environment should consider the effect of other natural and anthropogenic soil compounds and properties on prions, their adsorption kinetics as an important variable, as well as changes in PrP^CWD^ aggregation and conformation over time and upon binding to soil, specifically how these changes affect agent persistence, bioavailability and infectivity.

## Figures and Tables

**Figure 1 pathogens-12-00269-f001:**
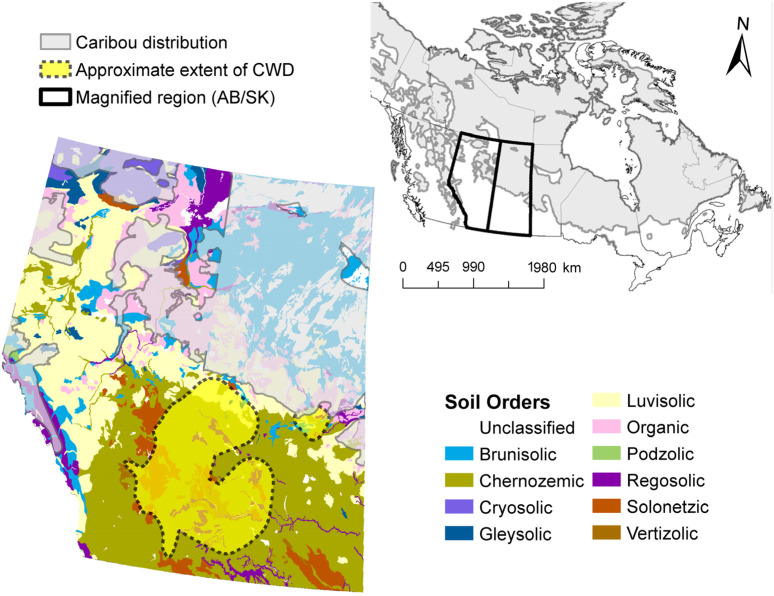
Soil order map of Western Canada (Alberta and Saskatchewan) with range of caribou (*Rangifer tarandus* spp.) and CWD-endemic region. (Soil order map source: Agriculture and Agri-Food Canada, 2010, v.3.1, accessed through ESRICanadaEd).

**Figure 2 pathogens-12-00269-f002:**
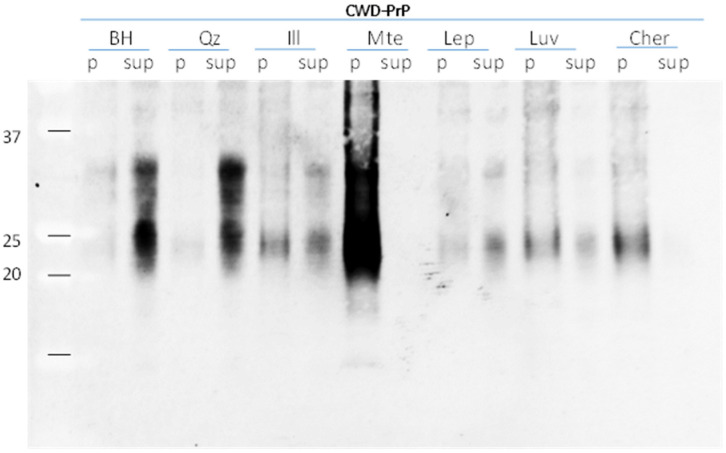
Binding capacity of soil minerals to WTD-CWD prions. Identical amounts of 10% infected deer brain homogenate (BH) were incubated with water (control) and 15 mg/mL suspension of soil minerals quartz (Qz), montmorillonite (Mte), Illite (Ill) and soils Leptosol (Lep), Luvisol (Luv) and Chernozem (Cher) overnight at 4 °C. Samples were fractionated through a 1M sucrose cushion to separate bound prions from unbound; pellet (p.; bound prions) and supernatant (sup; unbound prions) were analyzed by Western blot using mAb Bar224.

**Figure 3 pathogens-12-00269-f003:**
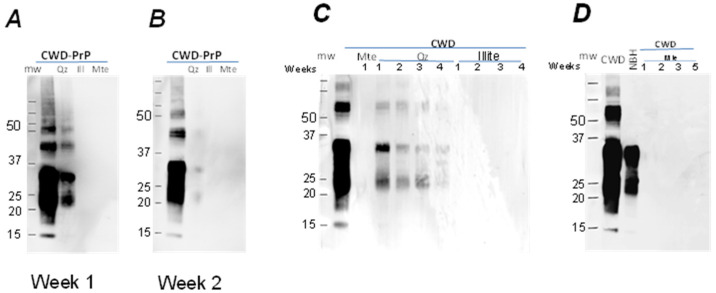
Leachate obtained at the first (**A**) and second (**B**) week and at weeks 1–4 (**C**,**D**) from columns with quartz (Qz), illite (Ill) with Mte (column experiment 1): Qz—pure quartz; Ill—illite (30%) + quartz; Mte—montmorillonite (30%) + quartz (70%). Approximately 5 mL of leachate was obtained from each column every week; 50 μL of leachate were boiled with 5 × Laemmlii buffer and analyzed by Western blot using mAb Bar224 (1:10,000). Initial amount of BH applied for the columns was used as a positive control (first lanes on each panel).

**Figure 4 pathogens-12-00269-f004:**
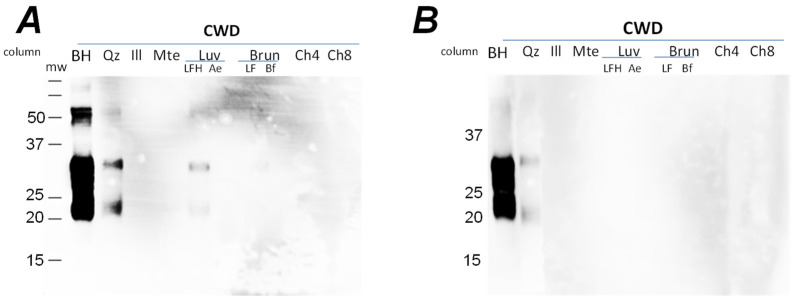
Leachate obtained at the first (**A**) and second (**B**) week from all 9 columns (column experiment 2): Qz—quartz; Ill—illite (50%) + quartz; Mte—montmorillonite (50%) + quartz; Luv—northern Luvisol; horizons LFH and Ae; Brun—mountain Brunisol; horizons LFH and Bf; Ch—Chernozems. Approximately 5 mL of leachate was obtained from each column; 50 μL of leachate was boiled at 100 °C with 5 × Laemmlii buffer and analyzed by Western blot using mAb Bar224 (1:10,000). Initial amount of BH applied for the columns was used as a positive control (first lanes on each panel).

**Figure 5 pathogens-12-00269-f005:**
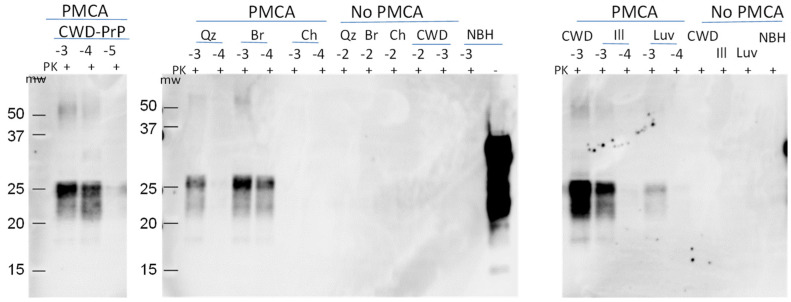
PMCA seeding activity of leachates from the columns. Second round of PMCA: the amplification of PrP^CWD^ in leachates obtained at the third week with quartz (Qz), illite (Ill), Brunisolic Bf horizon (Br), Luvisolic Ae horizon (Luv) and southern Chernozem (Ch) (experiment 2) columns. CWD-PrP (−3; −4; −5)—amplification control; Qz, Br, Ch, Ill, Luv—samples from column; NBH—uninfected brain homogenate diluted 1:10 in the PrP^C^ substrate; no PMCA control. The same samples were stored at 37 °C without sonication. The samples were PK digested (50 µg/µL) and analyzed by WB using mAb Bar224 (1:10,000).

**Figure 6 pathogens-12-00269-f006:**
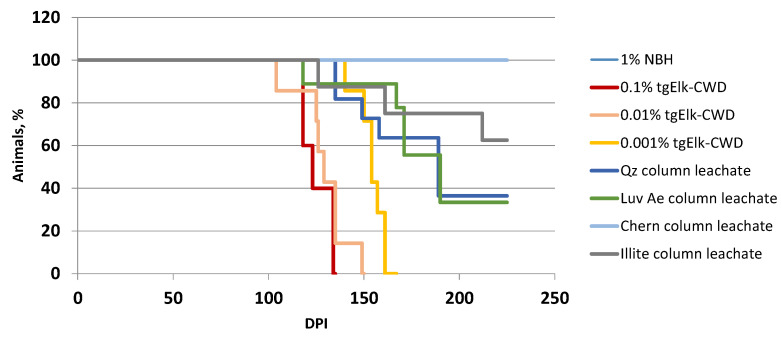
Infectivity of leachates obtained from the columns. Survival curve of tgElk mice i.c. inoculated with leachates obtained at the second week from columns with quartz (Qz), illite, Luvisolic Ae (Luv) and Chernozemic Ah (Cher) horizon. Dilution of infective tgElk CWD-BH (0.1%, 0.01%, 0.001%) was included as a control; NBH—uninfected tgElk BH.

**Figure 7 pathogens-12-00269-f007:**
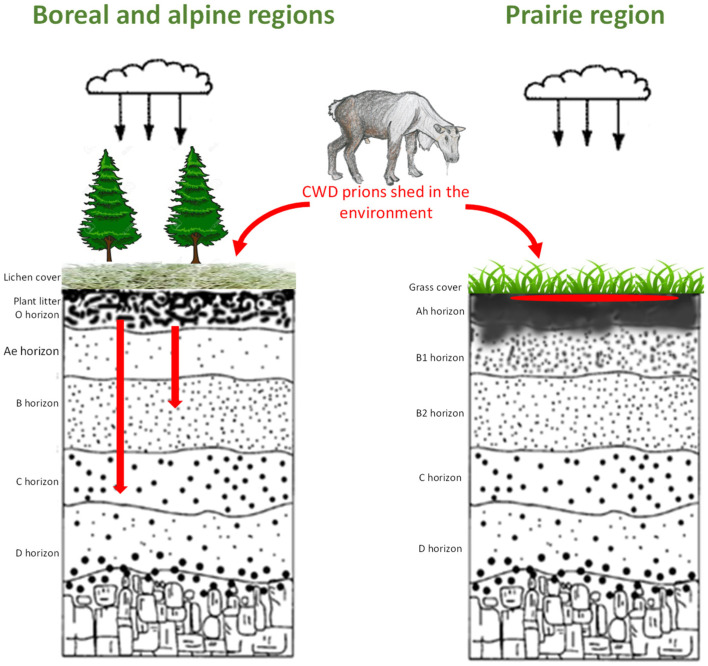
Prion fate in soils in different ecozones. In boreal and alpine/tundra regions, prions (PrP^CWD^ labeled as a red spot or arrows on the cartoon) that passed plant litter will move to lower soil horizons and become unavailable for cervids; in prairie regions, prions would be bound in upper soil horizons, remaining biologically available.

**Table 1 pathogens-12-00269-t001:** Properties of collected soil samples.

ID	Soil	Location	Horizon	pH	TOC, %	Humic Acid Content, g L^−1^	Texture	Mineralogy of Clay Fraction
Cher1	Orthic Dark Brown Chernozem	South-central region, Alberta, Canada	Ah *	7.9	3.9	15	Clay loam	Mte-Kte
Cher2	Orthic Black Chernozem	Central region, Leduc, Alberta, Canada	Ah	7.3	4	19	Silt loam	Mte-Kte
MBr	Gleyed Dystric Brunisol	Mountain region, Old Entrance, Alberta, Canada	LF ^¥^	4.8	38	1.5	N/A (org.hor.)	N/A (org.hor.)
			Bf ^#^	6.3	1.0	0.5	Loam	Mica-illite
NBr	Gleyed Eluviated Melanic Brunisol	Northern region, Hangingstone river, Alberta, Canada	LFH ^€^	4.0	35	3.8	N/A (org.hor.)	N/A (org.hor.)
			Ae ^£^	5.3	1.3	0.5	Silt loam	Mica-illite
NLuv	Podzolic Grey Luvisol	Northern region, Hangingstone river, Alberta, Canada	LFH ^€^	3.7	40	4.2	N/A (org.hor.)	N/A (org.hor.)
			Ae	5.5	1.1	0.2	Silt loam	Mica-illite
Lep	Leptosol	Nordfjella region, Norway	A	4.8	21	0.8	Silt sand	Vermiculate, quartz

* Ah—Surface mineral horizon enriched with organic matter; ^¥^ LF—Surface organic horizon consisting of relatively fresh plant litter (L), fermented litter (F); ^#^ Bf—Mineral horizon 10–50 cm below the surface enriched with iron or aluminum; ^€^ LFH—Surface organic horizon consisting of relatively fresh litter (L), fermented litter (F) and well-decomposed litter, humus (H); ^£^ Ae—Surface or near surface mineral horizon where clay and organic material has leached out.

**Table 2 pathogens-12-00269-t002:** Soil and mineral columns exposed to infected and uninfected brain homogenates (expose time).

Soil Columns	Infected BH	Uninfected BH
Quartz (Qz)	3 weeks and 6 weeks	6 weeks
Illite	3 weeks and 6 weeks	6 weeks
Mte	3 weeks and 6 weeks	6 weeks
Luvisolic LFH	3 weeks	--
Luvisolic Ae	3 weeks	--
Brunisolic LF	3 weeks	--
Brunisolic Bf	3 weeks	--
Chernozem 1	3 weeks	--
Chernozem 2	3 weeks	--

## Data Availability

All of the data are present in the manuscript.

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
