# Peer review of "Movement of Chronic Wasting Disease Prions in Prairie, Boreal and Alpine Soils"

_pathogens, 2023, doi:10.3390/pathogens12020269_

Round 1
Reviewer 1 Report
In this manuscript, Aiken and collaborators test the absortion of prions in different type of soils found in CWD endemic areas in North America and Noruega. As the authors clearly explain on introduction, this information could be important as some soils could act as "prion conservative", increasing the horizontal transfer of prions through feces, urine and other bodily fluids. To demonstrate this theory, the authors use different type of components present in different kind of soils and test its interaction with brain derived prions. This selection is based on bibliography and, in my opinion, is very clear and well explained. Lastly, the authors use real soils taken from different parts of North America and Norway to confirm the theory.
The results obtained are very clear and support the conclusions, evidencing the difference in prion absorbance and conservation on the surface or fast transport to lower layers, making them non available for cervid ingestion. The manuscript is very clear and direct, giving a lot of details about the different components present in different kind of soils and their possible interaction with PrPSc or PrPC according to the results obtained. Furthermore, the authors use sucrose cushion, PMCA and transgenic mouse bioassay to demonstrate the presence or abscence of prions. These techniques are highly consolidated and robust, proving the original hypothesis and supporting the final conclusions of the authors. However, there is some minor issues that should be adressed by authors before the final publication of the manuscript:
1-In figure 2 we can observe a clear difference in total quantity of PrP fibrils between Mte and the rest of the samples. As a control of strong interaction with prion fibrils, it is clear that the quantity of PrP fibrils on the pellet is bigger than in the supernatant. However, the total quantity of PrP (supernatant + pellet) looks higher in the Mte sample. Did the authors use the same ammount and the same brain homogenate in all samples? How is it possible that we observe a clear difference on total prion quantity (supernatant + pellet) between, for example, the Mte samples and the Cher sample?
2-In lines 263 to 267 the authors say: "A 20kDa protein mass is equivalent to 1.78 nm (min. radius) (Erickson, 2009), thus PrP might vary from 1.78 to 2.1 nm (37kDa). The PrP is, therefore, smaller than the inter-layer space of mte and could be adsorbed between the clay layers. The interlayer binding can explain the avid binding and strong association of PrP to mte". However, PrPSc structure, which is the real component of he prion fibrils that the authors are concerned, have a completely different structure and size . The authors should add some comment about this matter.
3-In lines 309 to 311 the authors say: "For the whole soils, PrPCWD migrated through the organic (LFH-type) horizon of Luvisolic soils and the pure quartz during the first week (Fig. 4), but was only detected in the quartz column leachate during the second (Fig. 4) and third weeks of the experiment." However, there is no clear evidence of the three week results. The authors reference to figure 4, but in figure 4 there is results of one and two weeks, but not from the third week. The authors should include this result at least on supplementary matherial or eliminate this comment.
Author Response
We thank the reviewers for their comments and suggestions. We are also pleased with reviewers’ enthusiasm regarding our manuscript. The questions and critiques raised are answered in this resubmission.
Reviewer 1
1-In figure 2 we can observe a clear difference in total quantity of PrP fibrils between Mte and the rest of the samples. As a control of strong interaction with prion fibrils, it is clear that the quantity of PrP fibrils on the pellet is bigger than in the supernatant. However, the total quantity of PrP (supernatant + pellet) looks higher in the Mte sample. Did the authors use the same ammount and the same brain homogenate in all samples? How is it possible that we observe a clear difference on total prion quantity (supernatant + pellet) between, for example, the Mte samples and the Cher sample?
Yes, we added same amount of BH to soil and mineral samples. From our experience, plastic tubes could also adsorb protein resulting in a decline intotal recovered prion protein. Mte has highest binding capacity for the prions compare to quartz and illite so we suggest that while mte adsorb entire amount of added PrP, in other tubes PrP could also be adsorbed to the tube walls and not recovered neither pellet, nor supernatant. It could explain difference in total PrP recovered from each tube.
2-In lines 263 to 267 the authors say: "A 20kDa protein mass is equivalent to 1.78 nm (min. radius) (Erickson, 2009), thus PrP might vary from 1.78 to 2.1 nm (37kDa). The PrP is, therefore, smaller than the inter-layer space of mte and could be adsorbed between the clay layers. The interlayer binding can explain the avid binding and strong association of PrP to mte". However, PrPSc structure, which is the real component of he prion fibrils that the authors are concerned, have a completely different structure and size . The authors should add some comment about this matter.
We agree that PrP has complex structure that cannot be simply described by size, here we using the common definition of PrP, a protein of molecular weight 33–35 kDa (Detlev Riesner, Biochemistry and structure of PrPC and PrPSc, British Medical Bulletin, Volume 66, Issue 1, June 2003, Pages 21–33, https://doi.org/10.1093/bmb/66.1.21), and this size might be comparable to interlayer space of minerals and “could be adsorbed between the clay layers”.
We modified the sentence in the manuscript, now it states: ” A 20kDa protein mass is equivalent to 1.78 nm (min. radius) (Erickson, 2009), thus PrPC might vary from 1.78 to 2.1 nm (37kDa). The unglycosylated PrP is, therefore, smaller than the inter-layer space of mte and some peripheral part of molecule could be adsorbed between the clay layers.”
3-In lines 309 to 311 the authors say: "For the whole soils, PrPCWD migrated through the organic (LFH-type) horizon of Luvisolic soils and the pure quartz during the first week (Fig. 4), but was only detected in the quartz column leachate during the second (Fig. 4) and third weeks of the experiment." However, there is no clear evidence of the three week results. The authors reference to figure 4, but in figure 4 there is results of one and two weeks, but not from the third week. The authors should include this result at least on supplementary matherial or eliminate this comment.
We agree and have added a blot showing leachates from columns for third week to the supplementary section – new figure S3, and also added reference to the text.
Reviewer 2 Report
The article by Alsu Kuznetsova et al., analyzes the movement of prions associated with Chronic Wasting Disease (CWD) in various types of soils, including prairie, boreal, and alpine soils. It is well-written and the results are both interesting and important. Some points need revision, mainly regarding the presentation of data in the figures:
Figure 1: Clarify the meaning of BH in the figure legend. Indicate that the molecular weight marker was loaded in the first lane and the molecular weight of the lowest band.
Figure 2: On each condition, the signal should be distributed between the pellet and the supernatant as the prions are either bound or unbound. It is unexpected that in some conditions low signal is observed while a high signal is observed in others, such as Mte pellet. If the signal is low or absent for some mineral conditions, where is the signal lost? The authors should discuss these results observed. Were molecular weight markers loaded in the first lane? Indicate the molecular weight of the bands. The last should be done in all western blot panels.
Figure 3: Again, were molecular weight markers loaded in the first lane? What has been loaded at the second lane on panels A, B and C?
Figure 4.A and 4.B: The first lane is marked as “column”. Is this column showing the molecular marker? What has been loaded at the second lane on panels A and B?
0.4 ml of BH was applied for each column and around 5 ml were obtained. Hence, the BH was diluted more than 10 times and no or very low PrP signal is detected. The concentration of the leachate proteins would allow comparing the results in a similar concentration range to obtain an informative result.
Figure 5. Again, were molecular weight markers loaded in the first lane of both the first and second immunoblots? Only abbreviations for quartz and illite were indicated in the figure legend.
Figure 6. Only the abbreviation for quartz was indicated in the figure legend. The survival curves for 1% NBH and for “Chern column leachate” share similar colors and both survival curves cannot be distinguished.
Table 1: HA term is not explained either in the footnote or in the text. Clarify footnotes/terms in Table 1. LFH¥ or LF¥?
Line 361 Fig. 6 could be figure S3?
Author Response
We thank the reviewers for their comments and suggestions. We are also pleased with reviewers’ enthusiasm regarding our manuscript. The questions and critiques raised are answered in this resubmission. Our responses in this cover letter and the changes in the main manuscript are highlighted for easy identification.
Reviewer 2
The article by Alsu Kuznetsova et al., analyzes the movement of prions associated with Chronic Wasting Disease (CWD) in various types of soils, including prairie, boreal, and alpine soils. It is well-written and the results are both interesting and important. Some points need revision, mainly regarding the presentation of data in the figures:
Figure 1: Clarify the meaning of BH in the figure legend. Indicate that the molecular weight marker was loaded in the first lane and the molecular weight of the lowest band.
We edited the figure and legend as requested
Figure 2: On each condition, the signal should be distributed between the pellet and the supernatant as the prions are either bound or unbound. It is unexpected that in some conditions low signal is observed while a high signal is observed in others, such as Mte pellet. If the signal is low or absent for some mineral conditions, where is the signal lost? The authors should discuss these results observed. Were molecular weight markers loaded in the first lane? Indicate the molecular weight of the bands. The last should be done in all western blot panels.
We edited the figure and text as requested. From our experience, plastic tubes could also adsorb protein and total recovered amount of prions is usually less that was added initially. It could also explain difference in total PrP recovered from each tube.
Figure 3: Again, were molecular weight markers loaded in the first lane? What has been loaded at the second lane on panels A, B and C?
We edited the figure as requested: yes, the first lane is a molecular marker, the second lanes on all panels are BH (positive control)
Figure 4.A and 4.B: The first lane is marked as “column”. Is this column showing the molecular marker? What has been loaded at the second lane on panels A and B?
0.4 ml of BH was applied for each column and around 5 ml were obtained. Hence, the BH was diluted more than 10 times and no or very low PrP signal is detected. The concentration of the leachate proteins would allow comparing the results in a similar concentration range to obtain an informative result.
We edited the figure as requested. The only quartz column showed that PrP signal passed through column – as a positive control (first lane) we used initial amount of BH applied for column and PrP amount passed through column was diluted that is clear on western blot.
Figure 5. Again, were molecular weight markers loaded in the first lane of both the first and second immunoblots? Only abbreviations for quartz and illite were indicated in the figure legend.
We edited the figure and legend as requested
Figure 6. Only the abbreviation for quartz was indicated in the figure legend. The survival curves for 1% NBH and for “Chern column leachate” share similar colors and both survival curves cannot be distinguished.
We edited the legend. The colours of survival curves for NBH and Chernozem are different but they are not distinguishable because they have exactly same trend – no animals showed clinical signs during the incubation.
Table 1: HA term is not explained either in the footnote or in the text. Clarify footnotes/terms in Table 1. LFH¥ or LF¥?
We edited the table.
Line 361 Fig. 6 could be figure S3?
Yes, thank you for pointing this out – we edited this.